# A New Soft-Clipping Discrete Beta GARCH Model and Its Application on Measles Infection

Huaping Chen 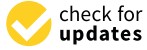

School of Mathematics and Statistics, Henan University, Kaifeng 475004, China; chenhp0107@henu.edu.cn

**Abstract:** In this paper, we develop a novel soft-clipping discrete beta GARCH (ScDBGARCH) model that provides an available method to model bounded time series with under-dispersion, equi-dispersion or over-dispersion. The new model not only allows positive dependence, but also negative dependence. The stochastic properties of the models are established, and these results are, in turn, used in the analysis of the asymptotic properties of the conditional maximum likelihood (CML) estimator of the new model. In addition, we apply the new model to measles infection to show its improved performance.

**Keywords:** discrete beta distribution; bounded time series; ScDBGARCH model; negative dependence; stochastic property

## 1. Introduction

More and more authors have underlined the importance and the common occurrence of the bounded integer-valued time series over more than two decades. McKenzie [1] proposed the binomial AR (BAR) model based on the binomial thinning operator to analyze the bounded integer-valued time series. To monitor bounded data (which increase at a certain point and then slowly decrease to the initial level), Weiß and Testik [2] further discussed the BAR model by constructing positive additive outliers; see Möller et al. [3] for its some extensions of zero inflation and Chen et al. [4] for its two types of innovative outliers. Kang et al. [5] proposed an extended binomial AR(1) model based on the generalized binomial thinning operator, which relaxes the independence assumption of the binomial thinning operator. To analyze bounded data with under-dispersion, equi-dispersion and over-dispersion, Chen et al. [6] first constructed the Conway–Maxwell–Poisson–binomial thinning operator based on the Conway–Maxwell–Poisson–binomial distribution [7], and then proposed the Conway–Maxwell–Poisson–binomial AR model. To accurately and flexibly capture the correlation structure between two random coefficients in the BAR process, Zhang et al. [8] proposed a new version of the BAR model by using the Farlie–Gumbel–Morgenstern copula, which allows both positive and negative correlations.

In fact, the volatility (especially the heteroscedasticity) is a reality for many important processes and cannot be described by the above models. Here, we take the number of districts with new cases of measles infection per week in the year 2016–2017 reported in $n = 38$ Germany's districts as an example and present its path in Figure 1, which shows that there seems to be more variation at the median of the time series, where the level also appears to be higher.

For this purpose, Weiß and Pollett [9] considered a linear binomial ARCH(1) model, which is generalized to the $p$th-order case by Ristić et al. [10]. Lee and Lee [11] further discussed a version of the linear binomial ARCH(1) model with a feedback mechanism. Chen et al. [12] proposed two classes of dynamic binomial ARCH models to model time series with a finite range. Chen et al. [13] generalized the binomial ARCH model to the beta-binomial GARCH model, which allows both the conditional and marginal binomial indices of dispersion to be greater than one, i.e., data with extra-binomial variation can be

more adequately captured than binomial GARCH-type models. See Liu et al. [14] for the bounded Poisson AR process and Liu et al. [15] for the novel category AR process.

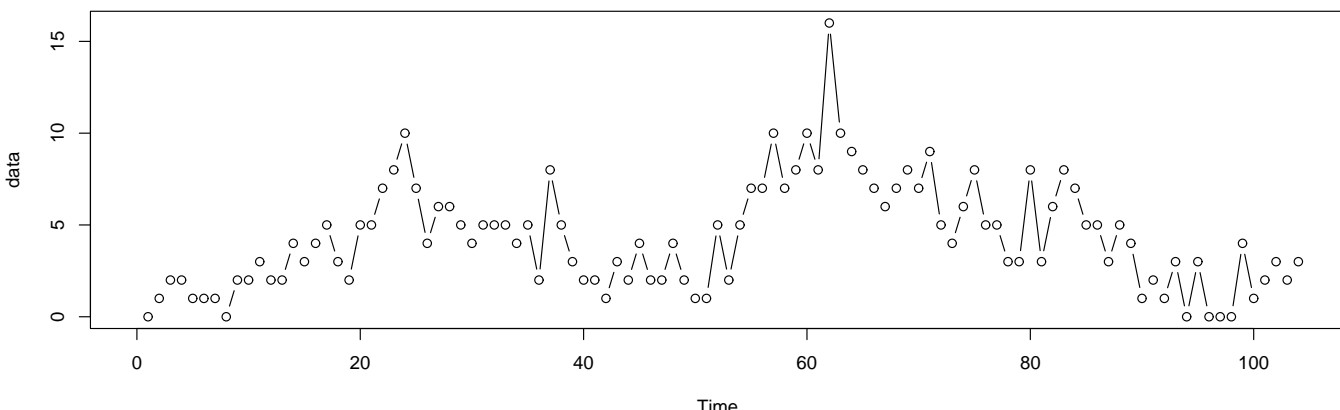

**Figure 1.** Path of the measles infection counts.

However, the negative ACF cannot be achieved by the above models. To resolve this dilemma, Weiß and Jahn [16], inspired by the softplus INGARCH model [17], proposed the soft-clipping BGARCH model based on the soft-clipping function [18], i.e.,

$$Sc_c(x) = c \log\left(\frac{1 + \exp(x/c)}{1 + \exp((x-1)/c)}\right), x \in (0,1), c > 0. \tag{1}$$

To further investigate the soft-clipping function, we give some example of the plot of $Sc_c(x)$ in Figure 2, when $c$ takes the value in $\{0.02, 0.05, 0.1, 0.3, 0.5, 0.7, 1, 1.5\}$, $\forall x \in (0,1)$. From Figure 2, $Sc_c(x)$ tends to a linear function when $c \to 0$.

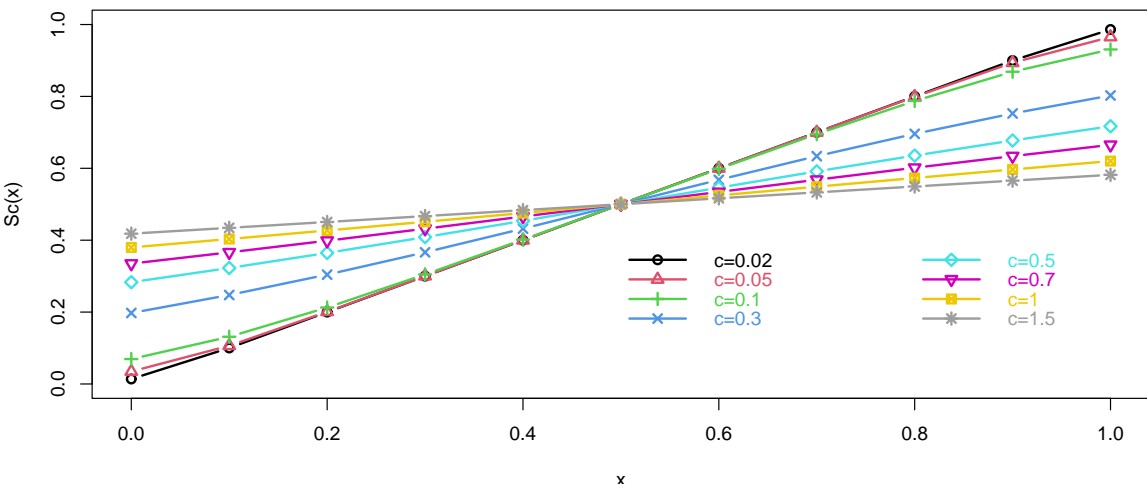

**Figure 2.** Plots of the soft-clipping function.

"Although the beta-binomial distribution is very flexible with respect to its shape, it is, to a large extent focused on dealing with data sets which appear, in some way, to arise from binomial distributions but which are in fact overdispersed", which was discussed by Turner [19]. Hence, another concern arises because beta-binomial distribution focuses on over-dispersion such that under-dispersed pseudo-binomial data sets (which are rare but do exist) cannot be analyzed by the beta-binomial GARCH-type models. To fill this gap, we proposed a new soft-clipping discrete beta GARCH (ScDBGARCH) model based on a re-scaled discrete beta binomial distribution. What is remarkable about the ScDBGARCH

model is that it not only can be fitted to under-dispersed data (besides over-dispersed bounded data), but also allows negative dependence (besides positive dependence).

It is worth mentioning that the realization of negative dependence for the ScDB-GARCH model is mainly due to the incorporated soft-clipping function. Another main contribution of this paper is that we establish the stochastic order of the discrete beta binomial distribution, and then discuss the stability property of the new model. In addition, we discuss the CML estimators and establish their asymptotic normality. Last but not least, we illustrate the availability and superiority in analyzing the count of districts with new cases of measles infection per week in the period of the year 2016–2017 reported in $n = 38$ of Germany's districts.

The paper is organized as follows. Section 2 first gives a brief review of the discrete beta distribution, then gives the definition of the soft-clipping discrete beta GARCH model and its stability properties. Conditional maximum likelihood estimation and their asymptotic properties are established in Section 3. Section 4 provides real data to show the effectiveness of the new model. Conclusions are made in Section 5. Appendix A presents some auxiliary results.

## 2. Model Formulation and Stability Properties

### 2.1. Discrete Beta Distribution

For the readers' convenience, we first give a brief review of the discrete beta distribution, which is introduced by Turner [19].

A random variable $X$ taking values in $\{n_{\text{bot}}, n_{\text{bot}} + 1, n_{\text{bot}} + 2, \ldots, n_{\text{top}}\}$ is said to follow a discrete beta distribution with parameters $(\alpha, \beta)$ if its probability mass function of $X$ takes the form

$$P(X = x | \alpha, \beta, n) = \frac{1}{Z(\alpha, \beta)} f\left(\frac{x - n_{\text{bot}} + 1}{n_{\text{top}} - n_{\text{bot}} + 2}\right), \forall x = n_{\text{bot}}, n_{\text{bot}} + 1, \ldots, n_{\text{top}}, \quad (2)$$

where

$$f(x) = \frac{1}{B(\alpha, \beta)} x^{\alpha - 1} (1 - x)^{\beta - 1}, \ B(\alpha, \beta) = \frac{\Gamma(\alpha)\Gamma(\beta)}{\Gamma(\alpha + \beta)}, \ Z(\alpha, \beta) = \sum_{x = n_{\text{bot}}}^{n_{\text{top}}} f\left(\frac{x - n_{\text{bot}} + 1}{n_{\text{top}} - n_{\text{bot}} + 2}\right),$$

where $n_{\text{top}} \in \mathbb{N}$ is the predetermined upper limit of the range and $n_{\text{bot}} = 0$ or $1$ is the predetermined lower limit of the range. For simplicity, we denote $X \sim \text{DB}^1(n_{\text{bot}}, n_{\text{top}}, \alpha, \beta)$.

Furthermore, the probability mass function (given in (2)) of $X$ can be rewritten as the exponential family form, i.e.,

$$P(X = x | \alpha, \beta) = h(x) \exp(\alpha T_1(x) + \beta T_2(x) - A(\alpha, \beta)), \quad (3)$$

where $A(\alpha, \beta) = \log\left(\sum_{x = n_{\text{bot}}}^{n_{\text{top}}} h(x) \exp\left(\alpha T_1(x) + \beta T_2(x)\right)\right), \ T_1(x) = \log\left(\frac{x - n_{\text{bot}} + 1}{n_{\text{top}} - n_{\text{bot}} + 2}\right),$

$T_2(x) = \log\left(\frac{n_{\text{top}} - x + 1}{n_{\text{top}} - n_{\text{bot}} + 2}\right), \ h(x) = \frac{(n_{\text{top}} - n_{\text{bot}} + 2)^2}{(x - n_{\text{bot}} + 1)(n_{\text{top}} - x + 1)}.$

In fact, $f(\cdot)$ involving in (2) is the probability density function of the beta distribution with parameters $\alpha$ and $\beta$. By Lemma A2 in Appendix A, one can obtain the mean, variance and BID of the $\text{DB}^1(n_{\text{bot}}, n_{\text{top}}, \alpha, \beta)$, if $n_{\text{bot}} = 0$ and $n_{\text{top}} \to \infty$. Similarly, the moments of $\text{DB}^1(1, n_{\text{top}}, \alpha, \beta)$ can be obtained if $n_{\text{top}} \to \infty$. It is worth mentioning that $\mu_b = \frac{\alpha}{\alpha + \beta}$ and $\sigma_b^2 = \frac{\alpha\beta}{(\alpha + \beta)^2(1 + \alpha + \beta)}$ (given in Lemma A2) are precisely the mean and variance of the beta distribution with parameters $\alpha$ and $\beta$. Hence, we consider a reparameterization of the discrete beta distribution given in (2) by setting $p = \alpha/(\alpha + \beta)$ and $\tau = \alpha + \beta$. For simplicity, we rewrite $\text{DB}^1(n_{\text{bot}}, n_{\text{top}}, \alpha, \beta)$ as $\text{DB}^2(n_{\text{bot}}, n_{\text{top}}, p, \tau)$.

Unfortunately, the specific range of BID for the $\text{DB}^2(n_{\text{bot}}, n_{\text{top}}, p, \phi)$ distribution cannot be obtained, except the case for $n_{\text{top}} \to +\infty$. To solve this dilemma, we give an example of

the BID in Figure 3 with $n := n_{\text{top}} \in (2, 4, 6, 8)$ and $n_{\text{bot}} = 0$, when $p$ and $\phi$ are varying from 0.1 to 0.9 with increment 0.1 and 0.1 to 8.1 with increment 0.1, respectively. See Figure 4 for $n_{\text{bot}} = 1$.

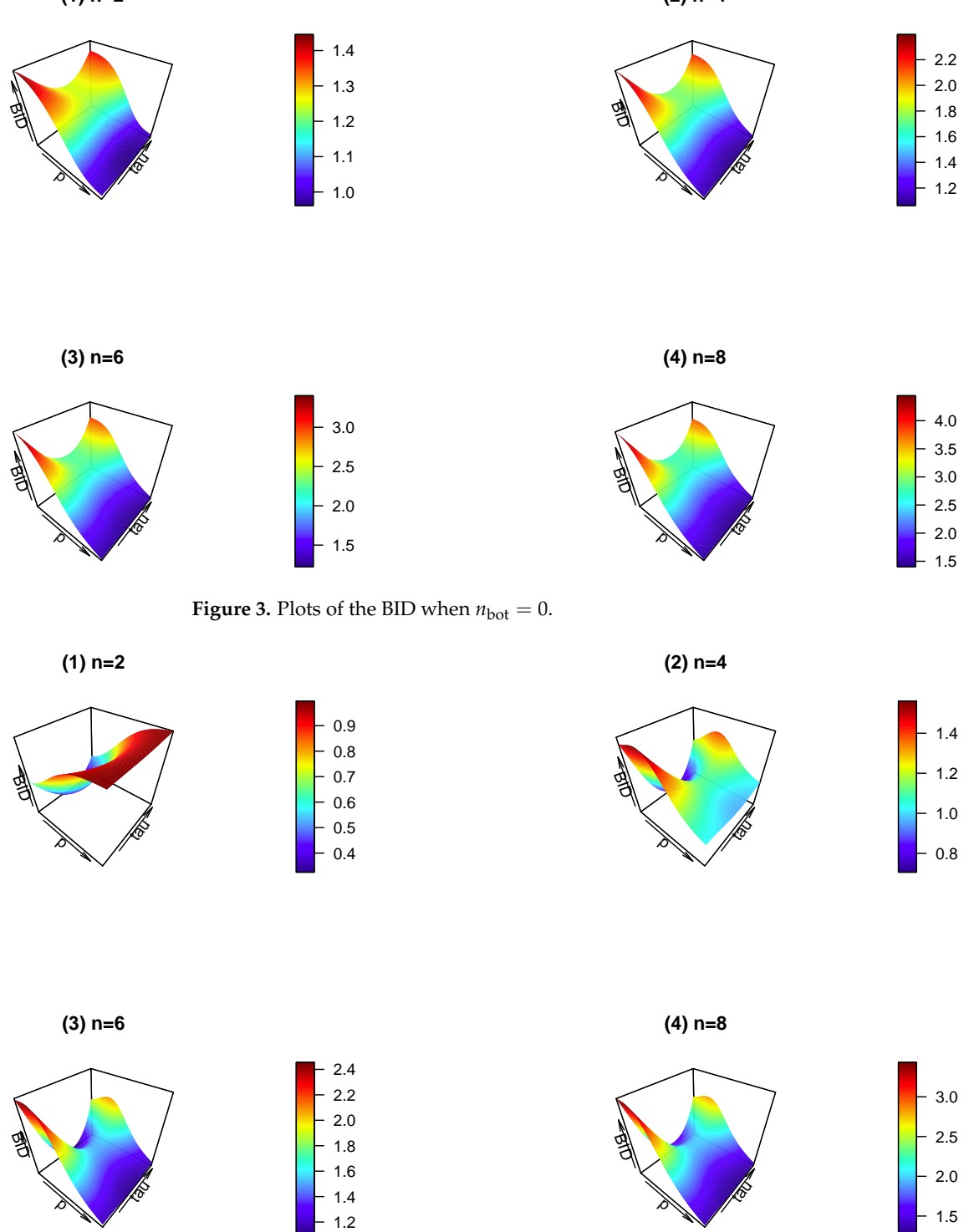

**Figure 3.** Plots of the BID when $n_{\text{bot}} = 0$.

**Figure 4.** Plots of the BID when $n_{\text{bot}} = 1$.

From Figure 3, we have the following observations. First, for given $\tau$, when $p \to 1$, the BID is decreasing but greater than 1, except for that of $n_{\text{top}} = 2$ (the BID is less than 1, if $p \to 1$). Second, when $p$ takes a small value, the BID takes the maximum if $\tau$ takes the

boundary value. Third, for the given $p$ and $\tau$, the BID tends to a greater value when $n_{\text{top}}$ is increasing.

From Figure 4, we have the following observations. First, for $n_{\text{top}} = 2$, the BID seems to be increasing but lower than 1 when $p \to 1$, $\forall \tau$. Second, for $n_{\text{top}} > 2$, if $\tau$ takes the non-boundary value, the BID seems to be increasing and then decreasing when $p \to 1$; otherwise, the BID seems to be decreasing. Third, for the given $p$ and $\tau$, the BID tends to a greater value when $n_{\text{top}}$ is increasing.

To sum up, the discrete beta distribution allows to model bounded data with under-dispersion, equi-dispersion and over-dispersion.

Similar to the statistical-order property of the one-parameter exponential family in [20], Proposition 1 illustrates that it does hold for the $\text{DB}^1$ distribution.

**Proposition 1.** *Suppose $X_i \sim \text{DB}^1(n_{\text{bot}}, n_{\text{top}}, \alpha_i, \beta_i)$, $\forall i = 1, 2$. If $\alpha_1 \leq \alpha_2$ and $\beta_1 \geq \beta_2$, then the following conclusions hold and are equivalent:*

*(1)* $X_1 \leq_{lr} X_2$,
*(2)* $p_1 \leq p_2$,
*where $p_i = \alpha_i / (\alpha_i + \beta_i)$, $\forall i = 1, 2$.*

**Proof.** (1) It is easy to see that $X_i$ exhibits the following probability density function

$$f_{X_i}(x) = \frac{1}{Z(\alpha_i, \beta_i)} \frac{1}{B(\alpha_i, \beta_i)} \left( \frac{x - n_{\text{bot}} + 1}{n_{\text{top}} - n_{\text{bot}} + 2} \right)^{\alpha_i - 1} \left( 1 - \frac{x - n_{\text{bot}} + 1}{n_{\text{top}} - n_{\text{bot}} + 2} \right)^{\beta_i - 1},$$

where $Z(\alpha_i, \beta_i) = \sum\limits_{x=0}^{n} \frac{1}{B(\alpha_i, \beta_i)} \left( \frac{x - n_{\text{bot}} + 1}{n_{\text{top}} - n_{\text{bot}} + 2} \right)^{\alpha_i - 1} \left( 1 - \frac{x - n_{\text{bot}} + 1}{n_{\text{top}} - n_{\text{bot}} + 2} \right)^{\beta_i - 1}$. Hence,

$$l(x) := \frac{f_{X_1}(x)}{f_{X_2}(x)} = \frac{Z(\alpha_2, \beta_2) B(\alpha_2, \beta_2)}{Z(\alpha_1, \beta_1) B(\alpha_1, \beta_1)} \left( \frac{x - n_{\text{bot}} + 1}{n_{\text{top}} - n_{\text{bot}} + 2} \right)^{\alpha_1 - \alpha_2} \left( 1 - \frac{x - n_{\text{bot}} + 1}{n_{\text{top}} - n_{\text{bot}} + 2} \right)^{\beta_1 - \beta_2}$$

$$\propto \left( \frac{x - n_{\text{bot}} + 1}{n_{\text{top}} - n_{\text{bot}} + 2} \right)^{\alpha_1 - \alpha_2} \left( 1 - \frac{x - n_{\text{bot}} + 1}{n_{\text{top}} - n_{\text{bot}} + 2} \right)^{\beta_1 - \beta_2}$$

and

$$l'(x) \propto (\alpha_1 - \alpha_2) \left( \frac{x - n_{\text{bot}} + 1}{n_{\text{top}} - n_{\text{bot}} + 2} \right)^{\alpha_1 - \alpha_2 - 1} \left( 1 - \frac{x - n_{\text{bot}} + 1}{n_{\text{top}} - n_{\text{bot}} + 2} \right)^{\beta_1 - \beta_2}$$

$$- (\beta_1 - \beta_2) \left( \frac{x - n_{\text{bot}} + 1}{n_{\text{top}} - n_{\text{bot}} + 2} \right)^{\alpha_1 - \alpha_2} \left( 1 - \frac{x - n_{\text{bot}} + 1}{n_{\text{top}} - n_{\text{bot}} + 2} \right)^{\beta_1 - \beta_2 - 1} \leq 0$$

with equality only if $\alpha_1 = \alpha_2$ and $\beta_1 = \beta_2$. Hence, $X_1 \leq_{lr} X_2$. Furthermore, $p_1 \leq p_2$ by Theorem 4.2 in Wang [21].

(2) Note that if $\alpha_1 \leq \alpha_2$ and $\beta_1 \geq \beta_2$,

$$\frac{1}{p_1} - \frac{1}{p_2} = \frac{\alpha_1 + \beta_1}{\alpha_1} - \frac{\alpha_2 + \beta_2}{\alpha_2} = \frac{\beta_1}{\alpha_1} - \frac{\beta_2}{\alpha_2} \geq 0.$$

Hence, $p_1 \leq p_2$, if $\alpha_1 \leq \alpha_2$ and $\beta_1 \geq \beta_2$, and vice versa. Therefore, $X_1 \leq_{lr} X_2$. The proof is end. $\square$

*2.2. Discrete Beta GARCH(1,1) Model with a Nearly Linear Structure*

Inspired by Weiß and Jahn [16] and $DB^2$ distribution, we give the definition of the ScDBGARCH(1,1) model by

$$\begin{cases} Z_t | \mathcal{F}_{t-1} \sim DB^2(n_{\text{bot}}, n_{\text{top}}, p_t, \tau), \\ p_t = Sc_c(w + \alpha_1 p_{t-1} + \beta_1 Z_{t-1}/n_{\text{top}}), \end{cases} \quad (4)$$

where $\mathcal{F}_t$ is the $\sigma$-field generated by $\{Z_t, p_t, t \in \mathbb{Z}\}$, $\tau > 0$, $|\alpha_1| < 1$, $|\beta_1| < 1$ and $|\alpha_1| + |\beta_1| < 1$, $Sc_c(x) = c \log \left( (1 + \exp(x/c))/(1 + \exp((x-1)/c)) \right), \forall x \in (0,1)$, $c > 0$, $n_{\text{bot}} = 0$ or 1 and $n_{\text{top}} \in \mathbb{N}$ is the predetermined upper limit of the range.

By (3) and (4), the conditional probability mass function of $\{Z_t\}$ takes the form

$$P(Z_t = z_t | \mathcal{F}_{t-1}) = h(z_t) \exp \left( p_t \tau T_1(z_t) + (1 - \tau) p_t T_2(z_t) - A(p_t \tau, (1 - \tau) p_t) \right), \quad (5)$$

where $h(z_t) = \dfrac{(n_{\text{top}} - n_{\text{bot}} + 2)^2}{(z_t - n_{\text{bot}} + 1)(n_{\text{top}} - z_t + 1)}$, $T_1(z_t) = \log \left( \dfrac{z_t - n_{\text{bot}} + 1}{n_{\text{top}} - n_{\text{bot}} + 2} \right)$, $T_2(z_t) = \log$
$\left( \dfrac{n_{\text{top}} - z_t + 1}{n_{\text{top}} - n_{\text{bot}} + 2} \right)$, $A(p_t \tau, (1 - \tau) p_t) = \log \left( \sum\limits_{i=n_{\text{bot}}}^{n_{\text{top}}} h(i) \exp(p_t \tau T_1(i) + (1 - \tau) p_t T_2(i)) \right)$.

Note that Proposition 1 presents that the new discrete beta distribution exhibits a statistical-order property, which is similar to the one-parameter exponential family in Davis and Liu [20]. Hence, a natural idea of the stability of the ScDBGARCH model is using the theory of the iterated random function approach [22] to construct the stability properties of the ScDBGARCH model. For this purpose, we first illustrate the stochastic order of the coupling process $\{Z_t, \lambda_t, t \in \mathbb{Z}\}$ given in (4), and then account for the moment property of $|Z_i - Z_j|$ ($\forall i \neq j$), which is essential to derive the stability of the proposed model.

**Proposition 2.** *If $\{Z_t, p_t, t \in \mathbb{Z}\}$ satisfies (4), then $Z_1 \leq_{lr} Z_2$, if $p_1 \leq_{lr} p_2$, where "lr" denotes the likelihood ratio.*

The result of Proposition 2 can be obtained by Proposition 1, $i = 1, 2$. We omit it.

**Proposition 3.** *For all $i = 1, 2$, if $Z_i \sim DB^2(n_{\text{bot}}, n_{\text{top}}, p_t, \tau)$ and $F_{\lambda_i}$ is the cumulative distribution function of $DB^2(n_{\text{bot}}, n_{\text{top}}, p_t, \tau)$ with $\mu_i = \sum_{z=n_{\text{bot}}}^{n_{\text{top}}} z P(Z_i = z)$ and $F_{\lambda_i}^{-1}(u) := \inf\{t \geq 0, F_{\mu_i}(t) \geq u\}$, then $E|Z_1 - Z_2| = |\lambda_1 - \lambda_2|$, where $u$ is a uniform random variable in $(0,1)$ and $Z_i = F_{\lambda_i}^{-1}(u)$.*

**Proof.** Denote $\lambda_i = \sum_{x=n_{\text{bot}}}^{n_{\text{top}}} x P(X_i = x)$ with $X_i \sim DB^1(n_{\text{bot}}, n_{\text{top}}, \alpha_i, \beta_i)$. Similar to the first item of Proposition 1, $\lambda_1 \leq_{lr} \lambda_2$, if $\alpha_1 \leq \alpha_2$, $\beta_1 \geq \beta_2$. Therefore, $Z_1 \leq_{lr} Z_2$ by Proposition 2, i.e., $Z_1 \leq_{st} Z_2$ and $F_{\lambda_1}^{-1}(t) \leq F_{\lambda_2}^{-1}(t), \forall t \in (0,1)$. Hence $E|Z_1 - Z_2| = E(Z_2 - Z_1) = \lambda_2 - \lambda_1 = |\lambda_1 - \lambda_2|$. Similarly, $E|Z_1 - Z_2| = E(Z_1 - Z_2) = |\lambda_1 - \lambda_2|$, if $\lambda_1 \geq \lambda_2$. Thus, $E|Z_1 - Z_2| = |\lambda_1 - \lambda_2|$. The proof is complete. □

In the following, we demonstrate that $Sc_c(\cdot)$ satisfies the contraction condition by using Lemma A1 in Appendix A, i.e., $\forall z_1, z_2 \geq 0 (z_1 \neq z_2)$, $p_1, p_2 \geq 0 (p_1 \neq p_2)$, there exist $\alpha$ and $\beta$ such that

$$\begin{cases} |Sc_c(w + \alpha_1 p_1 + \beta_1 \frac{z_1}{n_{\text{top}}}) - Sc_c(w + \alpha_1 p_2 + \beta_1 \frac{z_2}{n_{\text{top}}})| < |\alpha||p_1 - p_2| + |\beta||z_1 - z_2|, \\ Sc_c(w + \alpha_1 p_1 + \beta_1 z_1) \leq \alpha p_1 + \beta z_1 + Sc_c(w), \\ |Sc_c(w + \alpha_1 p_1 + \beta_1 z_1/n_{\text{top}}) - Sc_c(w + \alpha_1 p_2 + \beta_1 z_1/n_{\text{top}})| \leq |\alpha||p_1 - p_2|. \end{cases} \quad (6)$$

where $|\alpha_1| < 1$ and $|\beta_1| < 1$.

**Assumption 1.** *The parametric space* $\Theta = \{\boldsymbol{\theta} = (w, \alpha_1, \beta_1, \phi)^\top\}$ *is compact with* $w \in \mathbb{R}$, $0 < \phi < 1, |\alpha_1| < 1, |\beta_1| < 1$ *and* $|\alpha_1| + |\beta_1| < 1$.

**Theorem 1.** *Let* $\{Z_t, t \in \mathbb{Z}\}$ *satisfy* (4). *If Assumption* 1 *and the contraction condition* (6) *hold, then the following results hold:*

(1) *If* $\pi$ *is a stationary distribution and* $p_0 \sim \pi$ *is independent of* $p_0' \sim \pi$, *then* $\{p_t, t \in \mathbb{Z}\}$ *is geometric-moment contracting with unique stationary distribution* $\pi$ *and* $E_\pi p_1 < \infty$.

(2) *There exists a measurable function* $G_\infty : \mathcal{D}^\infty = \{(n_1, n_2, \ldots), n_i \in \mathcal{D}\} \longrightarrow \mathcal{D}$ *such that* $p_t \stackrel{a.s.}{=} G_\infty(Z_{t-1}, Z_{t-2}, \ldots)$, *i.e.,* $p_t$ *is* $\mathcal{F}_{t-1}$-*measurable, where* $\mathcal{D} = [0, n]$.

(3) *If* $\{p_t\}$ *starts from* $\pi$, *i.e.,* $p_0 \sim \pi$, *then* $\{Z_t\}$ *is a stationary time series. Furthermore,* $\{Z_t, p_t\}$ *is strictly stationary and ergodic.*

By Propositions 3 and (6), Theorem 1 can be proved in a similar way in Davis and Liu [20] and Chen et al. [13], and we omit it here.

It is worth mentioning that the incorporated soft-clipping function results in negative auto-regression, besides the positive auto-regression and over-dispersion. Unfortunately, because of the complexity of the discrete beta distribution, we have the closed forms of the auto-regressive coefficient. To get an idea about the abilities of the ScDBGARCH(1,1) model with $c = 0.01$ for explaining different autocorrelation structures, we present some ACF(2)-ACF(1) plots for the ScDBGARCH(1,1) model in Figures 5 and 6. To be precise, for given $n_{\text{top}} = 10$ and $n_{\text{bot}} = 0$ or 1, sample size $T = 200$ and $\tau = 1$, we let $\beta_1 = 0.05$ and $w = 0.5(1 - |\alpha_1| - |\beta_1|)$ with $\alpha_1$, varying from $-0.9$ to $0.9$ with an increment of $0.1$, and we compute the values of ACF(1), ACF(2) and plot them against each other.

From Figures 5 and 6, both negative ACF and non-negative ACF are allowed by the novel ScDBGARCH model, while negative ACF is rejected by the binomial GARCH-type models [10,12], i.e., the novel ScDBGARCH model is much more flexible than the classical binomial GARCH models with respect to the auto-regressive structure.

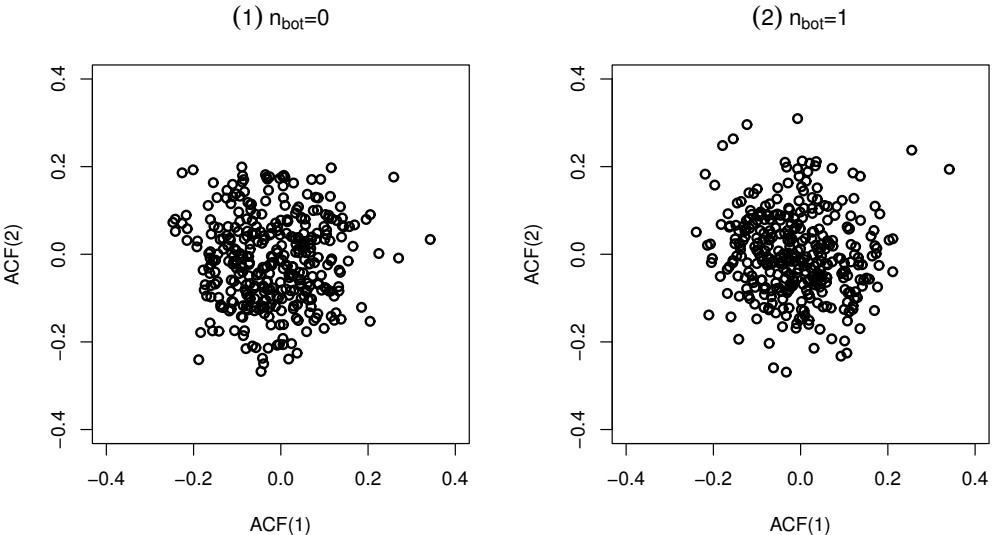

**Figure 5.** Plots of attainable pairs of ACF(2) against ACF(1) for $n_{\text{top}} = 10$ with $c = 0.01$.

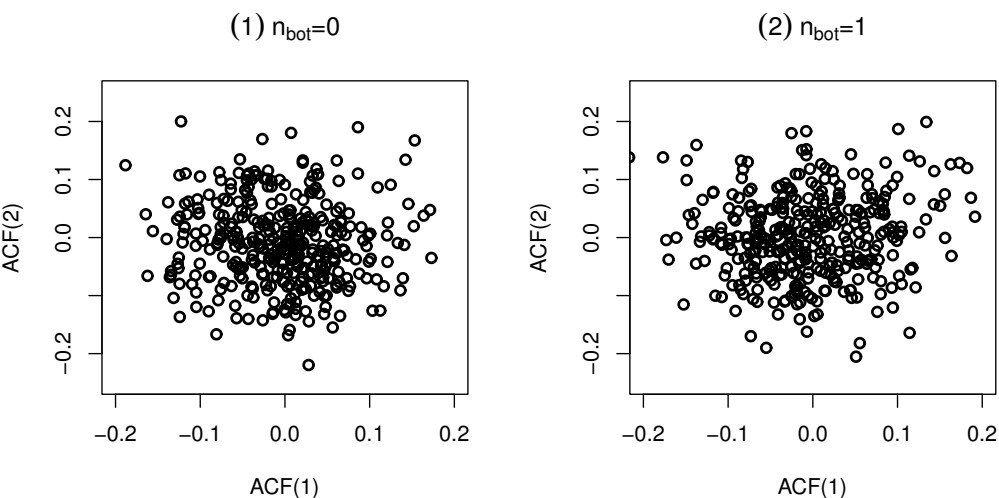

**Figure 6.** Plots of attainable pairs of ACF(2) against ACF(1) for $n_{\text{top}} = 2$ with $c = 0.01$.

To be honest, the merit of the model ScDBGARCH goes beyond allowing negative auto-regression, and also allowing under-dispersion. To account for the dispersion, we present the plots of the BID (in Figures 7 and 8) for the ScDBGARCH(1,1) model, for given $n_{\text{top}} = 10$ or 2 and $n_{\text{bot}} = 0$ or 1, sample size $T = 200$ and $\tau = 1$ when $\alpha_1$ is varying from $-0.9$ to $0.9$ with an increment 0.1, $\beta_1 = 0.05$ and $w = 0.5(1 - |\alpha_1| - |\beta_1|)$.

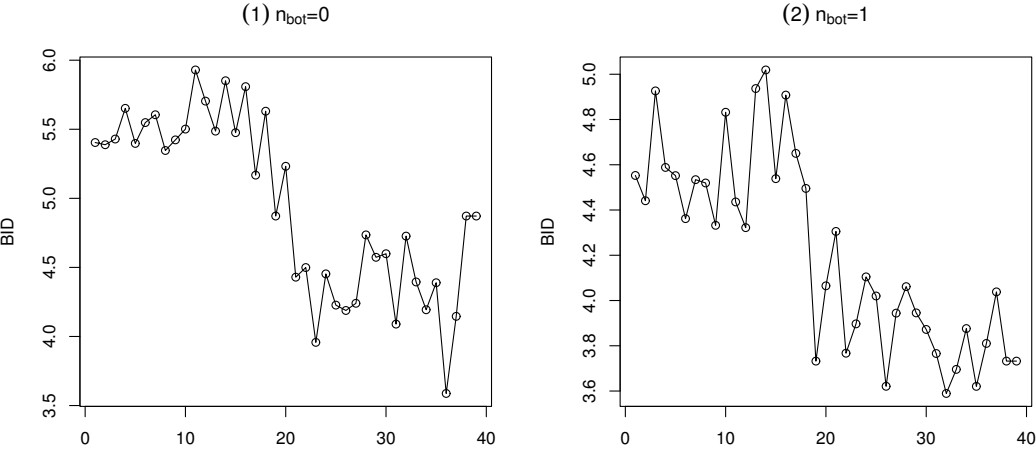

**Figure 7.** Plots of BID for $n_{\text{top}} = 10$ with $c = 0.01$.

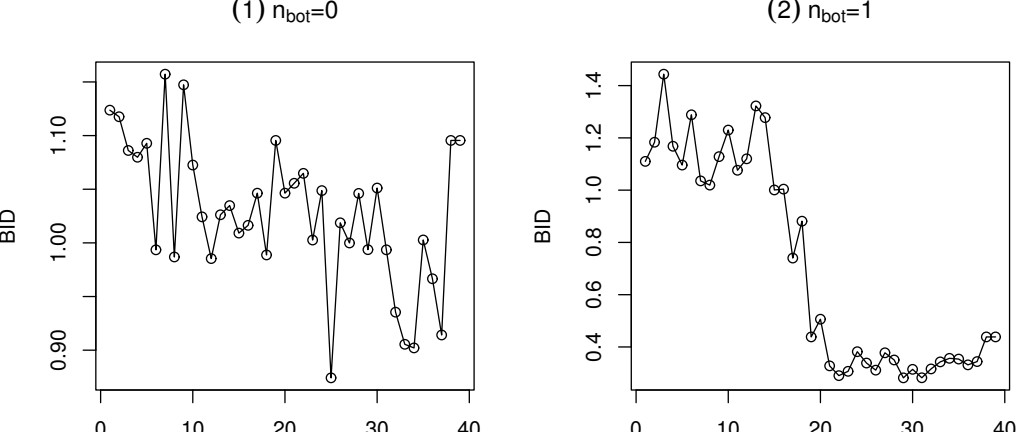

**Figure 8.** Plots of BID for $n_{\text{top}} = 2$ with $c = 0.01$.

From Figures [7] and [8], under-dispersion (besides over-dispersion) is allowed, especially for the ScDBGARCH model with a smaller $n_{\text{top}}$. Hence, the ScDBGARCH model provides an available way to analyze bounded integer-valued time series counts.

**Remark 1.** *Similar to the BGARCH(1,1) model [11] and the BBGARCH(1,1) model [13], we can define the following two models:*

- *Soft-clipping beta-binomial GARCH(1,1) model with*

$$Z_t|\mathcal{F}_{t-1} \sim \text{BB}(n, p_t, \phi), \ p_t = Sc_c(w + \alpha_1 p_{t-1} + \beta_1 Z_{t-1}/n), \tag{7}$$

*where $w \in \mathbb{R}$, $0 < \phi < 1$, $|\alpha_1| < 1$, $|\beta_1| < 1$ and $|\alpha_1| + |\beta_1| < 1$.*
*Obviously, this model, given in (7), is an example of the BBGARCH(1,1) model in [13]. For convenience, we recall it as the ScBBGARCH(1,1) model.*
- *Soft-clipping binomial GARCH(1,1) model [16] with*

$$Z_t|\mathcal{F}_{t-1} \sim \text{Bin}(n, p_t), \ p_t = Sc_c(w + \alpha_1 p_{t-1} + \beta_1 Z_{t-1}/n), \tag{8}$$

*where $w \in \mathbb{R}$, $0 < \phi < 1$, $|\alpha_1| < 1$, $|\beta_1| < 1$ and $|\alpha_1| + |\beta_1| < 1$.*
*Obviously, this model, given in (8), can be regarded as a further generation of the BARCH-type model; see [10–12]. For convenience, we recall it as the ScBGARCH(1,1) model.*

## 3. Parameter Estimation

In this section, we use the conditional maximum likelihood method to estimate the parameters involved in the ScDBGARCH(1,1) model and study their asymptotic behavior. Let $\theta = (w, \alpha, \beta, \tau)^{\top}$. Denote $n_{\text{top}}$ and $n_{\text{bot}}$ as the upper and lower ranges, and $T \in \mathbb{N}$ represents the size of the sample. $\{Z_0, Z_1, \ldots, Z_T\}$ is a realization of $\{Z_t\}$, which can be obtained by the following steps: First, we let $p_0 = Sc_c(w)$ and set a pre-run = 500, then generate $\{Z_0, p_1, Z_1, \cdots, p_{500}, Z_{500}\}$, where $p_t$ is obtained by (4) and $Z_t$ is generated by using rdb() function in the ddb package; see Turner [19] for more details. Second, we use $p_{500}$ as a new initial value of $p_t$ and rewrite it as $p_0$, then generate $\{Z_0, Z_1, Z_2, , Z_T\}$.

By (5), the conditional log-likelihood function of (4) can be written as

$$\begin{aligned}
\log L(\theta) &= \sum_{t=1}^{T} \log P(Z_t = z_t|\mathcal{F}_{t-1}) \\
&= \sum_{t=1}^{T} \log(h(z_t)) + \sum_{t=1}^{T} (\eta_1 T_1(z_t) + \eta_2 T_2(z_t) - A(\eta_1, \eta_2)),
\end{aligned} \tag{9}$$

where $h(z_t) = \dfrac{(n_{\text{top}} - n_{\text{bot}} + 2)^2}{(z_t - n_{\text{bot}} + 1)(n_{\text{top}} - z_t + 1)}$, $T_1(z_t) = \log\left(\dfrac{z_t - n_{\text{bot}} + 1}{n_{\text{top}} - n_{\text{bot}} + 2}\right)$, $\eta_1 = \tau p_t$,

$T_2(z_t) = \log\left(\dfrac{n_{\text{top}} - z_t + 1}{n_{\text{top}} - n_{\text{bot}} + 2}\right)$, $A(\eta_1, \eta_2) = \log\left(\sum_{i=n_{\text{bot}}}^{n_{\text{top}}} h(i) \exp(\eta_1 T_1(i) + \eta_2 T_2(i))\right)$ and

$\eta_2 = (1 - \tau)p_t$. Then the CML estimator $\hat{\theta}^{cml}$ is obtained by maximizing (9).

Note that $\sum_{t=1}^{T} \log(h(z_t))$ in (9) is a constant for a given sample. Hence, the conditional log-likelihood function given in (9) can be simplified and denoted as

$$\ell(\theta) = \sum_{t=1}^{T} l_t(\theta) = \sum_{t=1}^{T} (\eta_1 T_1(z_t) + \eta_2 T_2(z_t) - A(\eta_1, \eta_2)) \tag{10}$$

and $\hat{\boldsymbol{\theta}}^{cml}$ can be obtained by maximizing (10), i.e., $\hat{\boldsymbol{\theta}}^{cml}$ is a solution of the score equation

$$
\begin{aligned}
0 = \sum_{t=1}^{T} \frac{\partial l_t(\boldsymbol{\theta})}{\partial \boldsymbol{\theta}} &= \sum_{t=1}^{T} \left( T_1(z_t) \frac{\partial \eta_1}{\partial \boldsymbol{\theta}} - T_2(z_t) \frac{\partial \eta_2}{\partial \boldsymbol{\theta}} - \left( \frac{\partial A(\eta_1, \eta_2)}{\partial \eta_1} \frac{\partial \eta_1}{\partial \boldsymbol{\theta}} + \frac{\partial A(\eta_1, \eta_2)}{\partial \eta_2} \frac{\partial \eta_2}{\partial \boldsymbol{\theta}} \right) \right) \\
&= \sum_{t=1}^{T} \left( T_1(z_t) - \frac{\partial A(\eta_1, \eta_2)}{\partial \eta_1} \right) \frac{\partial \eta_1}{\partial \boldsymbol{\theta}} + \sum_{t=1}^{T} \left( T_2(z_t) - \frac{\partial A(\eta_1, \eta_2)}{\partial \eta_2} \right) \frac{\partial \eta_2}{\partial \boldsymbol{\theta}} \\
&= \sum_{t=1}^{T} \left( T_1(z_t) - A_1'(\eta_1, \eta_2) \right) \frac{\partial \eta_1}{\partial \boldsymbol{\theta}} + \sum_{t=1}^{T} \left( T_2(z_t) - A_2'(\eta_1, \eta_2) \right) \frac{\partial \eta_2}{\partial \boldsymbol{\theta}},
\end{aligned}
\tag{11}
$$

where $\eta_1 := \eta_1(\boldsymbol{\theta}) = \tau p_t$, $\eta_2 := \eta_2(\boldsymbol{\theta}) = p_t(1 - \tau)$, $p_t = Sc_c(u_t)$,

$$\partial A(\eta_1, \eta_2)/\partial \eta_1 := A_1'(\eta_1, \eta_2) = \sum_{i=n_{\text{bot}}}^{n_{\text{top}}} (h(i) \exp(\eta_1 T_1(i) + \eta_2 T_2(i)) T_1(i)) / B(\eta_1, \eta_2),$$

$$\partial A(\eta_1, \eta_2)/\partial \eta_2 := A_2'(\eta_1, \eta_2) = \sum_{i=n_{\text{bot}}}^{n_{\text{top}}} (h(i) \exp(\eta_1 T_1(i) + \eta_2 T_2(i)) T_2(i)) / B(\eta_1, \eta_2),$$

$$
\frac{\partial \eta_1}{\partial \boldsymbol{\theta}} = \begin{pmatrix} \tau Sc_c'(u_t) \\ \tau Sc_c'(u_t) p_{t-1} \\ \tau Sc_c'(u_t) z_{t-1}/n_{\text{top}} \\ Sc_c(u_t) \end{pmatrix}, \quad
\frac{\partial \eta_2}{\partial \boldsymbol{\theta}} = \begin{pmatrix} (1-\tau) Sc_c'(u_t) \\ (1-\tau) Sc_c'(u_t) p_{t-1} \\ (1-\tau) Sc_c'(u_t) z_{t-1}/n_{\text{top}} \\ -Sc_c(u_t) \end{pmatrix}
$$

with $A(\eta_1, \eta_2) = \log B(\eta_1, \eta_2)$, $B(\eta_1, \eta_2) = \sum_{i=n_{\text{bot}}}^{n_{\text{top}}} h(i) \exp(\eta_1 T_1(i) + \eta_2 T_2(i))$, $u_t = w + \alpha_1 p_{t-1} + \beta_1 z_{t-1}/n_{\text{top}}$ and $p_t = Sc_c(u_t)$.

Furthermore, the Hessian matrix (denoted as $H_T(\boldsymbol{\theta})$) for model (4) is obtained by further differentiation of the score equation, i.e., $H_T(\boldsymbol{\theta}) = -\sum_{t=1}^{T} \frac{\partial^2 l_t}{\partial \boldsymbol{\theta} \partial \boldsymbol{\theta}^\top}$ with $\frac{\partial^2 l_t}{\partial \boldsymbol{\theta} \partial \boldsymbol{\theta}^\top}$ equaling to

$$
(T_1 - A_1') \frac{\partial^2 \eta_1}{\partial \boldsymbol{\theta} \partial \boldsymbol{\theta}^\top} + (T_2 - A_2') \frac{\partial^2 \eta_2}{\partial \boldsymbol{\theta} \partial \boldsymbol{\theta}^\top} - A_{11}'' \frac{\partial \eta_1}{\partial \boldsymbol{\theta}} \frac{\partial \eta_1}{\partial \boldsymbol{\theta}^\top} - A_{22}'' \frac{\partial \eta_2}{\partial \boldsymbol{\theta}} \frac{\partial \eta_2}{\partial \boldsymbol{\theta}^\top} - (A_{12}'' + A_{21}'') \frac{\partial \eta_1}{\partial \boldsymbol{\theta}} \frac{\partial \eta_2}{\partial \boldsymbol{\theta}^\top},
$$

where $A_1' := A_1'(\eta_1, \eta_2)$, $A_2' := A_2'(\eta_1, \eta_2)$, $A_{ij}'' := A_{ij}''(\eta_1, \eta_2) = \partial A_i'(\eta_1, \eta_2)/\partial \eta_j$, $\forall i, j = 1, 2$ and

$$
\frac{\partial^2 \eta_1}{\partial \boldsymbol{\theta} \partial \boldsymbol{\theta}^\top} = \begin{pmatrix} \tau Sc_c'' & \tau Sc_c'' p_{t-1} & \tau Sc_c'' z_{t-1}/n_{\text{top}} & Sc_c' \\ \tau Sc_c'' p_{t-1} & \tau Sc_c'' p_{t-1}^2 & \tau Sc_c'' p_{t-1} z_{t-1}/n_{\text{top}} & p_{t-1} Sc_c' \\ \tau Sc_c'' z_{t-1}/n_{\text{top}} & \tau Sc_c'' p_{t-1} z_{t-1}/n_{\text{top}} & \tau Sc_c'' z_{t-1}^2/n_{\text{top}}^2 & Sc_c' z_{t-1}/n_{\text{top}} \\ Sc_c' & p_{t-1} Sc_c' & Sc_c' z_{t-1}/n_{\text{top}} & 0 \end{pmatrix},
$$

$$
\frac{\partial^2 \eta_2}{\partial \boldsymbol{\theta} \partial \boldsymbol{\theta}^\top} = \begin{pmatrix} (1-\tau) Sc_c'' & (1-\tau) Sc_c'' p_{t-1} & (1-\tau) Sc_c'' z_{t-1}/n_{\text{top}} & -Sc_c' \\ (1-\tau) Sc_c'' p_{t-1} & (1-\tau) Sc_c'' p_{t-1}^2 & (1-\tau) Sc_c'' p_{t-1} z_{t-1}/n_{\text{top}} & -p_{t-1} Sc_c' \\ (1-\tau) Sc_c'' z_{t-1}/n_{\text{top}} & (1-\tau) Sc_c'' p_{t-1} z_{t-1}/n_{\text{top}} & (1-\tau) Sc_c'' z_{t-1}^2/n_{\text{top}}^2 & -Sc_c' z_{t-1}/n_{\text{top}} \\ -Sc_c' & -p_{t-1} Sc_c' & -Sc_c' z_{t-1}/n_{\text{top}} & 0 \end{pmatrix}
$$

with $Sc_c' := Sc_c'(u_t)$ and $Sc_c'' := Sc_c''(u_t)$.

**Lemma 1.** *Denote $g(x, \eta_1, \eta_2) = T_1(x)\eta_1 + T_2(x)\eta_2 - A(\eta_1, \eta_2)$. For all $x \in \mathbb{R}$, $g(x, \eta_1, \eta_2) = g(x, \eta_1', \eta_2')$ if and only if $\eta_1 = \eta_1'$ and $\eta_2 = \eta_2'$, where $h(x) = \dfrac{(n_{\text{top}} - n_{\text{bot}} + 2)^2}{(x - n_{\text{bot}} + 1)(n_{\text{top}} - x + 1)}$, $T_1(x) = \log\left((x - n_{\text{bot}} + 1)/(n_{\text{top}} - n_{\text{bot}} + 2)\right)$, $T_2(x) = \log\left((n_{\text{top}} - x + 1)/(n_{\text{top}} - n_{\text{bot}} + 2)\right)$, $A(\eta_1, \eta_2) = \log\left(\sum_{i=n_{\text{bot}}}^{n_{\text{top}}} h(i) \exp(\eta_1 T_1(i) + \eta_2 T_2(i))\right)$, $n_{\text{bot}} = 0$ or $1$ and $n_{\text{top}}$ is considered a known quantity.*

**Proof.** Note that $g(x, \eta_1, \eta_2)$ is continuously differentiable; hence,

$$\frac{\partial g(x, \eta_1, \eta_2)}{\partial \eta_1} = T_1(x) - \frac{\sum_{i=0}^{n} h(i) T_1(i) \exp(\eta_1 T_1(i) + \eta_2 T_2(i))}{\sum_{i=0}^{n} h(i) \exp(\eta_1 T_1(i) + \eta_2 T_2(i))},$$

$$\frac{\partial g(x, \eta_1, \eta_2)}{\partial \eta_2} = T_2(x) - \frac{\sum_{i=0}^{n} h(i) T_2(i) \exp(\eta_1 T_1(i) + \eta_2 T_2(i))}{\sum_{i=0}^{n} h(i) \exp(\eta_1 T_1(i) + \eta_2 T_2(i))}.$$

Because $\left( \sum h(i) T_1(i) \exp(\eta_1 T_1(i) + \eta_2 T_2(i)) \right) / \left( \sum h(i) \exp(\eta_1 T_1(i) + \eta_2 T_2(i)) \right)$ is strictly increasing in terms of $\eta_1$ or $\eta_2$, so does for

$$\left( \sum h(i) T_2(i) \exp(\eta_1 T_1(i) + \eta_2 T_2(i)) \right) / \left( \sum h(i) \exp(\eta_1 T_1(i) + \eta_2 T_2(i)) \right).$$

Hence, $\partial g(x, \eta_1, \eta_2) / \partial \eta_1 = \partial g(x, \eta_1', \eta_2') / \partial \eta_2$ if and only if $\eta_1 = \eta_1'$ and $\eta_2 = \eta_2'$.

To sum up, $g(x, \eta_1, \eta_2) = g(x, \eta_1', \eta_2')$ if and only if $\eta_1 = \eta_1'$ and $\eta_2 = \eta_2'$, $\forall x \in \mathbb{R}$. $\square$

**Assumption 2.** *If there exists a* $t \geq 1$ *such that* $Z_t(\boldsymbol{\theta}_0) = Z_t(\boldsymbol{\theta})$, $P(z|\mathcal{F}_{-1})_{\boldsymbol{\theta}_0}$ *a.s., then* $\boldsymbol{\theta} = \boldsymbol{\theta}_0$, *where* $P(z|\mathcal{F}_{-1})_{\boldsymbol{\theta}_0} = P(Z_t = z | \mathcal{F}_{-1})_{\boldsymbol{\theta}_0}$ *is the probability measure under the true parameter* $\boldsymbol{\theta}_0$ *and* $\mathcal{F}_{-1}$.

Assumption 2 establishes the identification of the ScDBGARCH(1,1) model based on Lemma 1.

**Theorem 2.** *Let* $\{Z_t, t \in \mathbb{Z}\}$ *be a stationary and ergodic sequence with a finite range and its conditional mean process* $\{\lambda_t\}$ *satisfy* (4), *the contraction condition* (6). *If Assumptions 1 and 2 hold, then, as* $T \to \infty$, *we obtain the following results:*

(1)  *There exists an estimator* $\hat{\boldsymbol{\theta}}_2^{cml}$ *such that* $\hat{\boldsymbol{\theta}}^{cml} \xrightarrow{a.s.} \boldsymbol{\theta}$;

(2)  $\sqrt{T}(\hat{\boldsymbol{\theta}}^{cml} - \boldsymbol{\theta}) \xrightarrow{d} \mathcal{N}\left( \mathbf{0}, \, \boldsymbol{H}^{-1}(\boldsymbol{\theta}) \boldsymbol{I}(\boldsymbol{\theta}) \boldsymbol{H}^{-1}(\boldsymbol{\theta}) \right),$

   *where* $\boldsymbol{I}(\boldsymbol{\theta}) := E\left( \dfrac{\partial l_t(\boldsymbol{\theta})}{\partial \boldsymbol{\theta}} \dfrac{\partial l_t(\boldsymbol{\theta})}{\partial \boldsymbol{\theta}^\top} \right)$ *and* $\boldsymbol{H}(\boldsymbol{\theta}) := -E\left( \dfrac{\partial^2 l_t(\boldsymbol{\theta})}{\partial \boldsymbol{\theta} \partial \boldsymbol{\theta}^\top} \right)$.

The proof of Theorem 2 is similar to Theorem 4 in Chen et al. [12]. We omit it.

## 4. Real Data Example

In this section, we reconsider the number of districts with new cases of measles infection per week in the year 2016–2017 reported in $n = 38$ of Germany's districts. The dataset is taken from the "SurvStat" data (https://survstat.rki.de/Content/Query/Main.aspx) (accessed on 10 December 2022), which have been reported to the Robert Koch Institute by local and state health departments. See Figure 1 for its sample path.

By communication, the sample mean and variances are 4.3173 and 8.3546, respectively. The ACF and PACF plots are given in Figure 9, respectively.

Besides the ScDBGARCH(1,1) model, the ScBBGARCH(1,1) model given in (7) and the ScBGARCH(1,1) model given in (8) with $c = 0.01$, we also choose the following compared models:

- BARCH($p$) model [10] with

$$Z_t | \mathcal{F}_{t-1} \sim \text{Bin}(n, p_t), \, p_t = a_0 + \sum_{k=1}^{p} a_k Z_{t-k}/n, p = 1, 2;$$

- logit-BARCH($p$) model [12] with

$$Z_t | \mathcal{F}_{t-1} \sim \text{Bin}(n, p_t), \, \text{logit}(p_t) = \alpha_0 + \sum_{k=1}^{p} \text{logit}(\alpha_k) Z_{t-k}, p = 1, 2;$$

- score-BARCH(1) model [12] with

$$Z_t|\mathcal{F}_{t-1} \sim \text{Bin}(n, p_t), \text{logit}(p_t) = \alpha_0 + \alpha_1 \text{logit}(p_{t-1}) + \alpha_2 s_t, s_t = np_t - Z_t;$$

- BGARCH(1,1) model [11] with

$$Z_t|\mathcal{F}_{t-1} \sim \text{Bin}(n, p_t), p_t = \alpha_0 + \alpha_1 p_{t-1} + \alpha_2 Z_{t-1}/n;$$

- logit-BBGARCH(1,1) model [13] with $Z_t|\mathcal{F}_{t-1} \sim \text{BB}(n, p_t, \phi)$ and its mean process $\{\lambda_t\}$ satisfying $\text{logit}(\lambda_t) = w + \alpha_1 \text{logit}(\lambda_{t-1}) + \beta_1 Z_{t-1}$.

**(a)** **(b)**

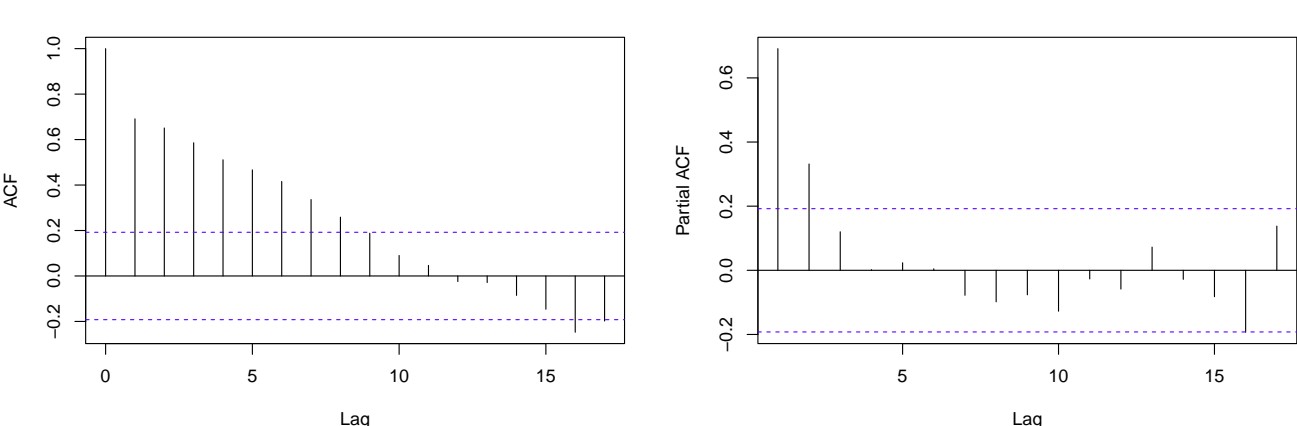

**Figure 9.** Measles infection's counts: (**a**) ACF, (**b**) PACF.

In the following, we use the above models to fit the measles infection's data by the CML method and compare their estimated standard error (SE), $-$log-likelihood ($-$log-lik), AIC and BIC, where SE is computed by Theorem 2 and $\hat{\phi} = 1/(1 + \hat{\tau})$. The CML estimates and approximated standard errors of parameters (including the fitted values of $-$log-lik, AIC and BIC) are summarized in Table 1.

From Table 1, we have the following observations. For the BARCH-type models, the BARCH(2) model with a linear transformation takes the smallest $-$log-lik, AIC and BIC. For the GARCH-type models, the ScDBGARCH(1,1) model takes the smallest $-$log-lik, AIC and BIC, followed by the ScBGARCH(1,1) model, which may be attributed to the merits of the soft-clipping function. For all compared models, the ScDBGARCH(1,1) model takes the smallest $-$log-lik, AIC and BIC. Hence, the ScDBGARCH(1,1) model is more suitable for the measles data.

To further check the adequacy of the ScDBGARCH(1,1) model, we analyze its Pearson residuals, which are defined by $e_t = (Z_t - \hat{\mu}_t)/\sqrt{\hat{\sigma}_t^2}$ with $\hat{\mu}_t = \sum_{z=0}^{n} Z_t P(Z_t = z|\mathcal{F}_{t-1})$ and $\hat{\sigma}_t^2 = \sum_{z=0}^{n} (z - \hat{\mu}_t)^2 P(Z_t = z|\mathcal{F}_{t-1})$. As discussed in Weiß [23], "for an adequate model, its fitted standardized Pearson residuals are expected to be uncorrelated with a mean about 0 and a variance about 1".

First, we calculate that the mean and variance of the Pearson residuals of the ScD-BGARCH(1,1) model are $-0.0059$ and $1.0107$, which implies that the ScDBGARCH(1,1) model demonstrates adequacy. Second, we give its residual analysis in Figure 10, which also shows that this model does rather well.

**Table 1.** Estimates and SEs in parentheses for the measles infection counts.

| Model | Estimates | | | | $-$log-lik | AIC | BIC |
|---|---|---|---|---|---|---|---|
| BARCH(1) | $\hat{a}_0$ 0.0367 (0.0071) | $\hat{a}_1$ 0.6844 (0.0651) | | | 212.6574 | 429.3148 | 434.6036 |
| BARCH(2) | $\hat{a}_0$ 0.0270 (0.0072) | $\hat{a}_1$ 0.4056 (0.0991) | $\hat{a}_2$ 0.3669 (0.1005) | | 204.8729 | 415.7457 | 423.6789 |
| logit-BARCH(1) | $\hat{a}_0$ $-2.8248$ (0.1002) | $\hat{a}_1$ 0.1608 (0.0161) | | | 215.9542 | 415.7457 | 423.5789 |
| logit-BARCH(2) | $\hat{a}_0$ $-2.9473$ (0.1087) | $\hat{a}_1$ 0.1042 (0.0221) | $\hat{a}_2$ 0.0827 (0.0220) | | 207.5645 | 421.1290 | 429.0622 |
| score-BARCH(1) | $\hat{\alpha}_0$ $-0.6178$ (0.1525) | $\hat{\alpha}_1$ 0.6777 (0.0751) | $\hat{\alpha}_2$ 0.1192 (0.0160) | | 213.1136 | 432.2272 | 440.1604 |
| BGARCH(1,1) | $\hat{a}_0$ 0.0332 (0.0087) | $\hat{a}_1$ 0.0175 (0.0263) | $\hat{a}_2$ 0.6923 (0.0675) | | 212.4199 | 430.8399 | 438.7730 |
| ScBGARCH(1,1) | $\hat{w}$ 0.2209 (0.2235) | $\hat{\alpha}_1$ 0.5123 (0.1002) | $\hat{\beta}_1$ 0.4292 (0.0836) | | 204.9207 | 415.8414 | 423.7746 |
| ScDBGARCH(1,1) | $\hat{w}$ 0.2130 (0.2297) | $\hat{\alpha}_1$ 0.4926 (0.0963) | $\hat{\beta}_1$ 0.4506 (0.0832) | $\hat{\phi}$ 0.0196 (0.0028) | 203.5400 | 415.0799 | 425.575 |
| ScBBGARCH(1,1) | $\hat{w}$ 0.3188 (0.3267) | $\hat{\alpha}_1$ 0.4946 (0.1333) | $\hat{\beta}_1$ 0.4401 (0.1116) | $\hat{\phi}$ 0.0202 (0.0174) | 211.9121 | 431.8242 | 442.4017 |
| logit-BBGARCH(1,1) | $\hat{w}$ $-1.7203$ (0.2657) | $\hat{\alpha}_1$ 0.4288 (0.0933) | $\hat{\beta}_1$ 0.1137 (0.0176) | $\hat{\phi}$ 0.0020 (0.0040) | 208.9151 | 425.8302 | 436.4078 |

Third, we consider the fitted values of the Ljung–Box test based on lags $k$ = 3, 5, 7, 9, 11, 13, and 15, including their $p$-values and their critical values ($\chi^2_{0.95}(k)$) with 0.05 confidence, and summarize them in Table 2.

**Table 2.** Values of the Ljung–Box test for the measles data.

| Lag $k$ | 3 | 5 | 7 | 9 | 11 | 13 | 15 |
|---|---|---|---|---|---|---|---|
| $p$-value | 0.7736 | 0.9519 | 0.9642 | 0.9916 | 0.9950 | 0.9800 | 0.9934 |
| $\chi^2_{0.95}(k)$ | 7.8147 | 11.0705 | 14.0671 | 16.9190 | 19.6751 | 22.3620 | 24.9958 |
| Ljung–Box statistic | 1.1144 | 1.1245 | 1.9191 | 1.9895 | 2.6083 | 4.7636 | 4.8430 |

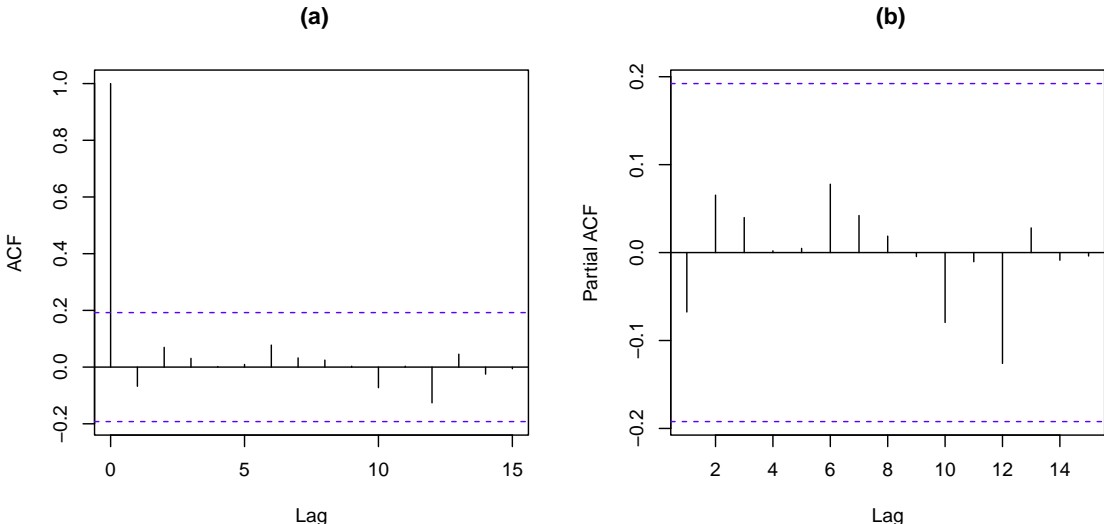

**Figure 10.** Pearson residual analysis: (**a**) ACF, (**b**) PACF.

Table 2 shows that all of the Ljung–Box statistics are less than the corresponding critical values, and the *p*-values are much greater than the significant level 0.05. Hence, both of them further illustrate the availability of the ScDBGARCH(1,1) model in analyzing the measles data. To sum up, the ScDBGARCH(1,1) model shows better performance in analyzing the measles data.

## 5. Concluding and Discussion

This paper considers a new and flexible soft-clipping discrete beta GARCH(1,1) model, which not only allows positive correlation, but also negative correlation, as well as under-dispersion, equi-dispersion and over-dispersion. We discuss some properties of the new model, the CML estimate of the parameters involved in the novel model, and the large-sample property of the CML estimate. The applicability and superior of the ScDBGARCH model are illustrated by a real data example.

Like linear binomial ARCH/GARCH-type models [10,11], logit binomial ARCH-type models [12] or beta-binomial GARCH-type models [13], the ScDBGARCH model is applicable to analyze stationary non-negative data with a finite range and will be invalid for data with some time trends. Two natural methods arise, and both of them deserve a detailed analysis in a future project.

One popular method is incorporated into the covariate processes when constructing a new model. Similar to the logit-BBGARCHX model [24] and the PARX model [25], one can establish a model with covariates, taking the ScDBGARCH(1,1) model as an example:

$$\begin{cases} Z_t | \mathcal{F}_{t-1} \sim \mathrm{DB}^2(nn_{\mathrm{top}}, n_{\mathrm{bot}}, p_t, \tau), \\ p_t = Sc_c(w + \alpha_1 p_{t-1} + \beta_1 Z_{t-1}/n_{\mathrm{top}} + f(\boldsymbol{X}_{t-1}, \boldsymbol{\gamma})), \end{cases}$$

where $n_{\mathrm{top}} \in \mathbb{N}$ is a predetermined upper limit of the range, $n_{\mathrm{bot}} = 0$ or 1 is a predetermined lower limit of the range, $\boldsymbol{X}_t = (X_{1t}, X_{2t}, \cdots, X_{dt})$ is a *d*-dimensional exogenous covariate vector, $\mathcal{F}_t$ is the $\sigma$-field generated by $\{Z_s, \lambda_s, \boldsymbol{X}_s, \forall s < t\}$, $f(\cdot, \boldsymbol{\gamma}) : \mathbb{R}^d \to \mathbb{R}$, $\boldsymbol{\gamma}$ is the additional parameter vector involved in $f(\cdot, \cdot)$, and $(w, \alpha_1, \beta_1, \tau)$ is the parameter vector with $\tau > 0, |\alpha_1| < 1, |\beta_1| < 1$ and $|\alpha_1| + |\beta_1| < 1$. When discussing the statistical property, an essential and unavoidable point is the specific form of $f(\cdot, \cdot)$. See Chen and Khamthong [26] for Markov-switching cases.

Specially, if the considered data have a periodic trend, one can consider a *s*-periodically distributed sequence $\{Z_t, t \in \mathbb{Z}\}$ and its mean process $\{\lambda_t\}$ satisfying (4), i.e., the *s*-periodicity of $\{Z_t, t \in \mathbb{Z}\}$ is understood in the sense that $Z_t \overset{d}{=} Z_{ks+\tau}$ for all $k, t \in \mathbb{Z}, \tau = 1, 2, \ldots, s$, where

$\stackrel{d}{=}$ denotes equality in distribution. To highlight the periodicity, one can consider the model by letting $t = ks + \tau, \forall k \in \mathbb{Z}, \forall \tau = 1, 2, \ldots, s$, and

$$
\begin{cases}
Z_{ks+\tau}|\mathcal{F}_{ks+\tau-1} \sim \mathrm{DB}^2(n, p_{ks+\tau}, \phi_{ks+\tau}), \\
\lambda_{ks+\tau} := (n+2)p_{ks+\tau} - 1 = S_c(w_{ks+\tau} + \alpha_{1,ks+\tau}\lambda_{ks+\tau-1} + \beta_{1,ks+\tau}Z_{ks+\tau-1}),
\end{cases}
$$

where $|\alpha_{1,\tau}| < 1$, $|\beta_{1,\tau}| < 1$ and $\prod_{\tau=1}^{s} |\alpha_{1,\tau}| + |\beta_{1,\tau}| < 1$. See Aknouche et al. [27] for a general periodic mixed Poisson autoregression.

The other popular method is to remove the time trend by using the difference method, but having a negative value emerge (besides non-negative bounded data), i.e., $\mathbb{Z}$-valued bounded data emerge. As far as we know, existing GARCH-type models are constructed by random rounding operators (see Liu and Yuan [28]), some $\mathbb{Z}$-valued discrete distributions (see Alomani et al. [29], Carallo et al. [30], Cui et al. [31]), difference of two independent non-negative INGARCH models (see Gonçalves and Mendes-Lopes [32]) and non-negative INGARCH models multiplying by some special $\mathbb{Z}$-valued discrete random variables (see Xu and Zhu [33]). However, they focus on $\mathbb{Z}$-valued data with infinite range and cannot apply to bounded data. Hence, a future project in term of the $\mathbb{Z}$-valued bounded data deserves to be considered.

In addition, as discussed in Chen et al. [6], the Conway–Maxwell–Poisson–binomial AR model shows better performance in analyzing bounded time series counts with under-dispersion, equi-dispersion and over-dispersion. A class of the Conway–Maxwell–Poisson–binomial GARCH model deserves to be considered to analyze volatility for integer-valued time series with a finite range. Similar to Bulla et al. [34] and Chen et al. [35], a signed Conway–Maxwell–Poisson–binomial (SCMPB) thinning operator and a bivariate INAR model based on the SCMPB thinning operator also deserve to be considered to analyze bivariate dependent time series with finite ranges.

**Funding:** Chen's work is funded by Natural Science Foundation of Henan Province (No. 222300420127) and Postdoctoral research in Henan Province (No. 202103051).

**Data Availability Statement:** The number of districts with new cases of measles infection per week in the year 2016–2017 reported in $n = 38$ Germany's districts is taken from the "SurvStat" (https://survstat.rki.de/Content/Query/Main.aspx) on 12 December 2019.

**Acknowledgments:** The author thanks the Editor-in-Chief and the anonymous referees for the valuable comments and suggestions that resulted in a substantial improvement of this paper. We acknowledge the constructive suggestions from Fukang Zhu of Jilin University on the work.

**Conflicts of Interest:** The author declares no conflict of interest.

## Abbreviations

The following abbreviations are used in this manuscript:

| | |
|---|---|
| $|x|$ | absolute of $x$, $x \in \mathbb{R}$; |
| lr | likelihood ratio; |
| $\leq_{st}$ | stochastic small; |
| $\stackrel{d}{=}$ | equality in distribution. |
| $\stackrel{a.s.}{\rightarrow}$ | almost surely convergence; |
| $\stackrel{d}{\rightarrow}$ | convergence in distribution. |

## Appendix A. Auxiliary Results

**Lemma A1.** *Let* $Sc_c(x) = c \log \dfrac{1 + \exp(x/c)}{1 + \exp((x-1)/c)}, \forall x \in \mathbb{R}, c > 0$. *Then,*

*(1)* $Sc_c'(x) = \dfrac{\exp(x/c)}{1 + \exp(x/c)} - \dfrac{\exp((x-1)/c)}{1 + \exp((x-1)/c)}$ *and* $|S_c'(x)| \leq 1/2$;

(2) $\forall x_1 \in \mathbb{R}, x_2 \in \mathbb{R}$ and $x_1 \neq x_2$, $|Sc_c(x_2) - Sc_c(x_1)| \leq \frac{1}{2}|x_2 - x_1|$;

(3) $Sc_c''(x) = \dfrac{\exp(x/c)}{c(1 + \exp(x/c))^2} - \dfrac{\exp((x-1)/c)}{c(1 + \exp((x-1)/c))^2}$ and $|Sc_c''(x)| \leq \dfrac{1}{2c}$;

(4) $Sc_c'''(x) = \dfrac{\exp(x/c)(\exp(x/c) - 1)}{c^2(1 + \exp((x-1)/c))^3} - \dfrac{\exp(x - 1/c)(\exp((x-1)/c) - 1)}{c^2(1 + \exp((x-1)/c))^3}$ and

$|Sc_c'''(x)| \leq \dfrac{1}{4c^2}$.

**Proof.** (1) Because $Sc_c(x)$ is a continuously differentiable function in $\mathbb{R}$, $Sc_c'(x)$ exists and

$$Sc_c'(x) = \frac{\exp(x/c)}{1 + \exp(x/c)} - \frac{\exp((x-1)/c)}{1 + \exp((x-1)/c)}$$

and

$$|Sc_c'(x)| \leq \left|\frac{\exp(x/c)}{1 + \exp(x/c)}\right| + \left|\frac{\exp((x-1)/c)}{1 + \exp((x-1)/c)}\right| \leq \frac{1}{4} + \frac{1}{4} = \frac{1}{2}$$

by Lemma 4 in [12].

(2) By using the mean value theorem, there exists at least one point $\delta \in (x_1, x_2), \forall x_1 \neq x_2$, such that

$$Sc_c(x_2) - Sc_c(x_1) = Sc_c'(\delta)(x_2 - x_1),$$

where $Sc_c'(\delta) = \dfrac{\exp(\xi/c)}{1 + \exp(\delta/c)} - \dfrac{\exp((\delta - 1)/c)}{1 + \exp((\delta - 1)/c)}$. Hence, $|Sc_c'(\delta)| \leq 1/2$ and $|Sc_c(x_2) - S_c(x_1)| \leq \dfrac{1}{2}|x_2 - x_1|, \forall x_1 \neq x_2$.

(3) According to item (1), $Sc_c'(x)$ is a continuously differentiable function in $\mathbb{R}$, thus $Sc_c''(x)$ exists and $Sc_c''(x) = \dfrac{\exp(x/c)}{c(1 + \exp(x/c))^2} - \dfrac{\exp((x-1)/c)}{c(1 + \exp((x-1)/c))^2}$. Furthermore,

$$|Sc_c''(x)| \leq \left|\frac{\exp(x/c)}{c(1 + \exp(x/c))^2}\right| + \left|\frac{\exp((x-1)/c)}{c(1 + \exp((x-1)/c))^2}\right| \leq 1/(4c) + 1/(4c) = 1/(2c)$$

by $(a + b)^2 \geq 4ab, \forall a \in \mathbb{R}, \forall b \in \mathbb{R}$.

(4) By (3), $Sc_c''(x)$ is a continuously differentiable function in $\mathbb{R}$, and thus, $Sc_c'''(x)$ exists and

$$Sc_c'''(x) = \frac{\exp(x/c)(\exp(x/c) - 1)}{c^2(1 + \exp(x/c))^3} - \frac{\exp((x-1)/c)(\exp((x-1)/c) - 1)}{c^2(1 + \exp((x-1)/c))^3}.$$

Furthermore, by using Lemma 4 in [12], we obtain

$$|Sc_c'''(x)| \leq \left|\frac{\exp(x/c)(\exp(x/c) - 1)}{c^2(1 + \exp(x/c))^3}\right| + \left|\frac{\exp((x-1)/c)(\exp((x-1)c) - 1)}{c^2(1 + \exp((x-1)/c))^3}\right|$$

$$\leq \frac{1}{c^2}\left|\frac{\exp(2x/c)}{(1 + \exp(x/c))^3}\right| + \frac{1}{c^2}\left|\frac{\exp(2(x-1)/c)}{(1 + \exp((x-1)/c))^3}\right|$$

$$\leq \frac{1}{4c^2}.$$

The proof is complete. $\square$

**Lemma A2.** *Let* $X \sim \mathrm{DB}^1(n_{\mathrm{bot}}, n_{\mathrm{top}}, \alpha, \beta)$ *with* $n_{\mathrm{bot}} = 0$ *and* $n_{\mathrm{top}} = n$. *If* $n \to +\infty$, *then*

(1). $E(X) \approx (n+2)\dfrac{\alpha}{\alpha+\beta} - 1 = (n+2)\mu_b - 1,$

(2). $\text{Var}(X) \approx (n+2)^2\sigma_b^2,$

(3). $\text{BID} = \dfrac{n\text{Var}(X)}{EX(n-EX)} \approx \dfrac{n\phi(n+2)^2 p(1-p)}{(n+2)^2 p(1-p) - (n+1)} \begin{cases} > 1, & \text{if } p(1-p) > \dfrac{n+1}{(1-n\phi)(n+2)^2}, \\ = 1, & \text{if } p(1-p) = \dfrac{n+1}{(1-n\phi)(n+2)^2}, \\ < 1, & \text{if } p(1-p) < \dfrac{n+1}{(1-n\phi)(n+2)^2}, \end{cases}$

*where $\mu_b = p = \alpha/(\alpha+\beta)$ and $\sigma_b^2 = \phi\mu_b(1-\mu_b)$ with $\phi = 1/(1+\alpha+\beta)$.*

**Proof.** By (2), we compute that

$$
\begin{aligned}
E(X) &= \frac{1}{Z(\alpha,\beta)}\sum_{x=0}^{n} xf\left(\frac{x+1}{n+2}\right) = \frac{(n+2)^2}{Z(\alpha,\beta)}\sum_{x=0}^{n}\left(\frac{x+1}{n+2} - \frac{1}{n+2}\right)f\left(\frac{x+1}{n+2}\right)\frac{1}{n+2} \\
&= \frac{(n+2)^2}{Z(\alpha,\beta)}\sum_{x=0}^{n}\frac{x+1}{n+2}f\left(\frac{x+1}{n+2}\right)\frac{1}{n+2} - \frac{(n+2)^2}{Z(\alpha,\beta)}\sum_{x=0}^{n}\frac{1}{n+2}f\left(\frac{x+1}{n+2}\right)\frac{1}{n+2} \\
&\approx \frac{(n+2)^2}{Z(\alpha,\beta)}\int_0^1 xf(x)dx - \frac{n+2}{Z(\alpha,\beta)}\int_0^1 f(x)dx \\
&\approx \frac{(n+2)^2}{n+2}\int_0^1 xf(x)dx - \frac{n+2}{n+2}\int_0^1 f(x)dx \\
&\approx (n+2)\frac{\alpha}{\alpha+\beta} - 1 = (n+2)\mu_b - 1,
\end{aligned}
$$

$$
\begin{aligned}
E(X^2) &= \frac{1}{Z(\alpha,\beta)}\sum_{x=0}^{n} x^2 f\left(\frac{x+1}{n+2}\right) \\
&= \frac{(n+2)^3}{Z(\alpha,\beta)}\sum_{x=0}^{n}\left(\frac{(x+1)^2}{(n+2)^2} - \frac{2(x+1)}{(n+2)^2} + \frac{1}{(n+2)^2}\right)f\left(\frac{x+1}{n+2}\right)\frac{1}{n+2} \\
&= \frac{(n+2)^3}{Z(\alpha,\beta)}\sum_{x=0}^{n}\left(\frac{x+1}{n+2}\right)^2 f\left(\frac{x+1}{n+2}\right)\frac{1}{n+2} - \frac{2(n+2)^2}{Z(\alpha,\beta)}\sum_{x=0}^{n}\frac{x+1}{n+2}f\left(\frac{x+1}{n+2}\right)\frac{1}{n+2} \\
&\quad + \frac{n+2}{Z(\alpha,\beta)}\sum_{x=0}^{n} f\left(\frac{x+1}{n+2}\right)\frac{1}{n+2} \\
&= \frac{(n+2)^3}{Z(\alpha,\beta)}\left(\int_0^1 x^2 f(x)dx - \left(\int_0^1 xf(x)dx\right)^2\right) + \frac{(n+2)^3}{Z(\alpha,\beta)}\left(\int_0^1 xf(x)dx\right)^2 \\
&\quad - \frac{2(n+2)^2}{Z(\alpha,\beta)}\int_0^1 xf(x)dx + \frac{n+2}{Z(\alpha,\beta)}\int_0^1 f(x)dx \\
&\approx (n+2)^2\frac{\alpha\beta}{(\alpha+\beta)^2(1+\alpha+\beta)} + (n+2)^2\frac{\alpha^2}{(\alpha+\beta)^2} - 2(n+2)\frac{\alpha}{\alpha+\beta} + 1 \\
&= (n+2)^2\sigma_b^2 + (n+2)^2\mu_b^2 - 2(n+2)\mu_b + 1,
\end{aligned}
$$

where $\sigma_b^2 = \dfrac{\alpha\beta}{(\alpha+\beta)^2(1+\alpha+\beta)}$ and $\mu_b = \dfrac{\alpha}{\alpha+\beta}$. Hence,

$$
\begin{aligned}
\text{Var}(X) &= E(X^2) - (EX)^2 \approx (n+2)^2\sigma_b^2 + (n+2)^2\mu_b^2 - 2(n+2)\mu_b + 1 - ((n+2)\mu_b - 1)^2 \\
&= (n+2)^2\sigma_b^2.
\end{aligned}
$$

Hence, the binomial index of dispersion (BID) of $X$ satisfies

$$\text{BID} = \frac{n\text{Var}(X)}{EX(n - EX)} \approx \frac{n\phi(n+2)^2 p(1-p)}{(n+2)^2 p(1-p) - (n+1)} \begin{cases} > 1, & \text{if } p(1-p) > \dfrac{n+1}{(1-n\phi)(n+2)^2}, \\ = 1, & \text{if } p(1-p) = \dfrac{n+1}{(1-n\phi)(n+2)^2}, \\ < 1, & \text{if } p(1-p) < \dfrac{n+1}{(1-n\phi)(n+2)^2}. \end{cases}$$

The proof is complete. $\square$

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
