# Peer review of "A New Soft-Clipping Discrete Beta GARCH Model and Its Application on Measles Infection"

_stats, doi:10.3390/stats6010018_

Round 1
Reviewer 1 Report
see pdf.

Author Response
Reply to Reviewer #1
Special thanks for your helpful suggestions and comments. We have made a complete revision which we hope to meet with your approval. Below is our point-to-point reply. Your comments are in italic.
1.There are some English mistakes, for example,
(1) “negative dependence” should be written as “Negative dependence”, and ”.)” in “Stochastic property.)” should be removed.
(2) It seems like that $\phi$ in line 79 denotes $\tau$, If so, please modify, else illustrate it clearly;
(3) $\alpha$ and $\beta$ in (6) should be modified as $\alpha_1$ and
$\beta_1$...
Answer: Thanks for your detailed and helpful suggestion. Now, we correct them, which are highlighted in the manuscript.
2. There are some problems in residual analysis in Figure 6, because there is no reason why Pearson residuals should be normally distributed (see also Möller et al. (2018), Zhang and Wang (2022), Yang et al. (2023), etc.); Thus, the lower panel in Figure 8 is not required.
Answer: You are right, now, the lower panel in Figure 8 is deleted.
3. Is the ScDBGARCH model identifiability of the model? If so, please elaborate it below Assumption 2, else, it arise some question about the asymptotic property of the parameters involving in ScDBGARCH model.
Answer: This is a good question. The ScDBGARCH model is identifiability, which can be obtained by Assumption 2 and Lemma 1. We are sorry that we forgot illustrating it, now, we correct it, see pages 10-11.
4. Lemma A2 has been not referred. Please elaborate.
Answer: Thanks for your detailed and helpful suggestion. Lemma A2 is used to account for the identifiability of the ScDBGARCH model, and we are sorry that we forgot illustrating it. Now, we move this Lemma to Section 3, i.e., Lemma 1(see page 10), which is necessary to illustrate the the identifiability of the ScDBGARCH model.

Author Response
Reply to Reviewer #2
Thank you very much for your attention to our manuscript and your professional comments. In the revision, we re-correct and rephrase those written in poor English. In addition, we point-to-point reply to those being not understandable and questionable. Below is our point-to-point reply. Your comments are in italic.
1. English proofreading is recommended. The grammar and sentence structure errors affect the understanding of the manuscript, such as
- Keywords, “negative dependence” should be written as “Negative dependence”, ”.)” in “Stochastic property.)” should be removed;
- Line 22, what does [?] denote as? Please rephrase;
- Line 42, “classes” should be modified by “class”;
- Line 43, “models” should be modified by “model”;
- Line 64, it seems better if the sentence “if the probability mass function of X takes the form” is written as “if its probability mass function takes the form ”;
- Line 78, “cases” should be modified as “case”;
- Line 79, what’s $\phi$? By the Figures 1 and 2, it seems that $\phi$ denotes $\tau$, If so, please modify, otherwise rephrase it clearly;
- $\alpha$ and $\beta$ in (6) should be modified as $\alpha_1$ and $\beta_1$;
Answer: Thanks for your detailed and helpful suggestion. Now, we correct them, which are highlighted in the manuscript.
2. The random number of the new DB distribution is missing. The author shall provide an algorithm about how to generate the random data.
Answer: This is a good suggestion. Now, we add it, see page 9.
3. The Lemma and its proof seem inconsistent, please rephrase.
Answer: Thanks for your detailed and helpful suggestion. Now, we write Lemmas 2 and 3 as Propositions 2 and 3, and the correct their proofs, see page 3.
4. What’s the purpose of Assumption 2 and Lemma A2? It seems better to illustrate it clearly.
Answer: This is a good suggestion. Now, we add it, see page 10.
5. The writing of the contents in lines 174 and 176 is less organized page layout. Please readjust.
Answer: Thanks for your detailed and helpful suggestion. Now, we correct them, see page 12.
6. The author considers the DB1 and DB2 distributions, while the binomial-discrete Poisson-lindley distribution is also an important model (the following article for detail), please point out this in the “Concluding remarks”.
[1] Chesneau C , Bakouch H S , Akdoan Y , et al. The Binomial-Discrete Poisson-Lindley Model: Modeling and Applications to Count Regression[J]. Commun. Math. Res, 2022(038-001).
Answer: This is a good suggestion, but the DB distribution is different from the Bin-PL distribution because the former takes value from $\{n_{\rm bot}, n_{\rm bot}+1, …, n_{\rm top}\}$, while the latter takes value from $\{0,1,2,…\}$. Hence, we do not refer this reference in the manuscript.
